# Multiple photofluorochromic luminogens via catalyst-free alkene oxidative cleavage photoreaction for dynamic 4D codes encryption

Lin Lu[1,2,3], Bo Wu[3], Xinyuan He[4], Fen Zhao[3], Xing Feng [5], Dong Wang [1], Zijie Qiu [3], Ting Han [1] ✉, Zheng Zhao [3] ✉ & Ben Zhong Tang [3,4] ✉

Controllable photofluorochromic systems with high contrast and multicolor in both solutions and solid states are ideal candidates for the development of dynamic artificial intelligence. However, it is still challenging to realize multiple photochromism within one single molecule, not to mention good controllability. Herein, we report an aggregation-induced emission luminogen TPE-2MO2NT that undergoes oxidation cleavage upon light irradiation and is accompanied by tunable multicolor emission from orange to blue with time-dependence. The photocleavage mechanism revealed that the self-generation of reactive oxidants driving the catalyst-free oxidative cleavage process. A comprehensive analysis of TPE-2MO2NT and other comparative molecules demonstrates that the TPE-2MO2NT molecular scaffold can be easily modified and extended. Further, the multicolor microenvironmental controllability of TPE-2MO2NT photoreaction within polymer matrices enables the fabrication of dynamic fluorescence images and 4D information codes, providing strategies for advanced controllable information encryption.

In nature, many animals can control their appearance in response to external stimuli. For example, chameleons possess the capacity to change their appearance color to match surroundings for hide or camouflage[1,2]. Learning from nature, scientists are motivated to design intelligent photofluorochromic systems, which can change their emission intensity or emission color in response to light irradiation. The high sensitivity and non-contact feature of light endow photofluorochromic materials with significant potential in advanced fields such as information storage, anticounterfeiting, and biological applications[3–10]. The photofluorochromic phenomenon of luminescent materials mostly originates from photoreactions, including photoisomerization, photocyclization, photodimerization, and photocleavage. Based on different photoreactions, abundant photofluorochromic materials have been reported by introducing typical photoresponsive skeletons such as azobenzene, dithienylethene, spiropyran, and *ortho*-nitrobenzyl group[11–18]. However, the reported photofluorochromic materials often suffer from low fluorescence contrast and single fluorescence response (intensity or color

[1]Center for AIE Research, Shenzhen Key Laboratory of Polymer Science and Technology, Guangdong Research Center for Interfacial Engineering of Functional Materials, College of Materials Science and Engineering, Shenzhen University, Shenzhen 518060, China. [2]College of Physics and Optoelectronic Engineering, Shenzhen University, Shenzhen 518060, China. [3]School of Science and Engineering, Shenzhen Institute of Aggregate Science and Technology, The Chinese University of Hong Kong, Shenzhen (CUHK-Shenzhen), Guangdong 518172, China. [4]Department of Chemistry, Hong Kong Branch of Chinese National Engineering Research Center for Tissue Restoration and Reconstruction, The Hong Kong University of Science and Technology, Clear Water Bay, Kowloon, Hong Kong 999077, China. [5]School of Material and Energy, Guangdong University of Technology, Guangzhou 510006, China. ✉e-mail: hanting@szu.edu.cn; zhaozheng@cuhk.edu.cn; tangbenz@cuhk.edu.cn

change), which limit their potential applications as intelligent materials[19–22]. Thus, the construction of a high contrast, controllable, and multicolor photofluorochromic system is of great significance not only for the exploration of photoreactions but also for the development of intelligent materials for advanced technology.

C=C double bond is a key structural component of conjugated functional materials. Oxidative cleavage of C=C double bond of alkene derivatives is an important reaction to produce ketones and aldehydes. However, traditional oxidative cleavage reactions often require the use of ozone or alternative oxidants, such as $KMnO_4$, $PhIO/HBF_4$, $OsO_4$, and $CrO_2Cl_2$, most of which are toxic and costly[23–26]. To improve oxidative cleavage, transition metal catalysts, photocatalytic methods, and nitroarenes oxygen transfer reagents have been developed[27–32]. For example, oxygen in the presence of external photocatalysts has been utilized for oxidative cleavage of activated alkenes mildly[33]. However, external catalysts required in these systems may either increase the cost or generate waste, which is not favorable for practical applications. In this regard, the development of catalyst-free oxidative cleavage of alkenes is significant but has rarely been reported[34]. Thus, developing photoreactions in the absence of external photocatalysts has attracted increasing attention but presents significant challenges.

On the other hand, oxidative cleavage of π-conjugated compounds breaks the π-conjugation of the molecular backbone, which would significantly influence the electron delocalization of the π-system and thus change the luminescent properties, suggesting that the oxidative cleavage reaction of alkenes potentially could be employed to design smart photofluorochromic systems. However, there are basically no studies on photofluorochromism based on the oxidative cleavage reaction of alkenes, which may be ascribed to the challenge of achieving efficient alkene-containing luminescent system with both facile oxidative cleavage capability and luminescence response[35]. Therefore, the development of alkene alkene-containing luminescent system that could be oxidatively cleaved without external additives may open a window for the design of photofluorochromic materials.

Because conjugated planar luminophores mostly showed aggregation-caused quenching (ACQ) effect upon aggregation, many developed photoresponsive luminescent materials showed their limitation when applied in aggregate/solid state. Constructing luminogens with aggregation-induced emission (AIE) effect and photoresponsive feature have been proven as an efficient strategy to construct excellent photofluorochromic materials since their photofluorochromic property could be well kept in aggregate/solid state without emission quenching[36–48]. Furthermore, some AIE luminogens emissions exhibited high sensitivity to the microenvironments of their aggregate states and thus could realize multiple photochromism within one single molecule. However, those reported AIE-active multiple photofluorochromic materials were still rather limited[49–51]. The integration of AIE luminogen and catalyst-free oxidative cleavage photoreaction might have the chance to endow AIE-active photofluorochromic materials with enriched luminescence behavior.

In this work, by integrating the design of AIE, twisted intramolecular charge transfer (TICT), reactive oxygen species (ROS) generation system, and photoinduced oxidative cleavage of alkene, we designed the multiple photofluorochromic AIE system based on alkene oxidative cleavage photoreaction. TPE-2MO2NT integrates three functionalities into the molecular design: luminescent unit for observing fluorescence change; ROS generation unit for producing reactive oxidant; and activated alkene for oxidizing cleavage (Fig. 1). The electron-donating and electron-withdrawing structure could reduce the energy level gap and facilitate intersystem crossing (ISC) processes, enabling the formation of highly reactive oxygen species (ROS) when reacting with $O_2$[52]. As activated alkene, TPE-2MO2NT modified by electron-donating methoxy groups and electron-withdrawing naphthalimide units could be easily oxidized by the self-produced ROS

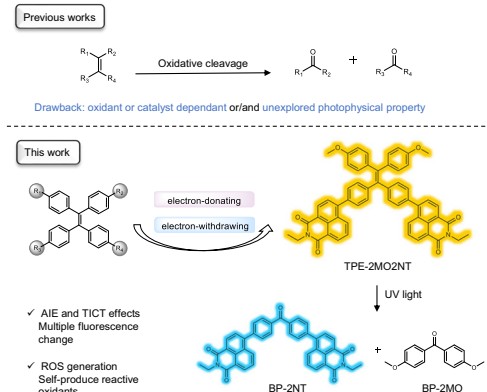

**Fig. 1 | Examples of oxidative cleavage of alkenes in reported works and the catalyst-free oxidation of TPE-2MO2NT in this work.** TPE-2MO2NT is modified by two electron-donating methoxy groups and two electron-withdrawing naphthalimide units and produced two photoproducts BP-2NT and BP-2MO. AIE: aggregation-induced emission; TICT: twisted intramolecular charge transfer; ROS: reactive oxygen species.

under light irradiation. The broken of C=C bond will destroy the original conjugation of TPE-2MO2NT to generate the photoconversion system based on BP-2NT and TPE-2MO2NT with tunable multiple fluorescence changes upon light irradiation. Accordingly, dynamic fluorescence imaging and time-gated 4D codes were successfully fabricated, showing the great potential of the multiple photofluorochromic AIEgen in advanced information storage and encryption.

## Results

### Synthesis and Photophysical Properties
TPE-2MO2NT was synthesized according to the synthetic route in Supplementary Fig. S1, and the structure of TPE-2MO2NT and all of the intermediates have been well characterized (Supplementary Figs. 2–7). The absorption and photoluminescent (PL) spectra were measured to systematically study its photophysical properties. As shown in Supplementary Fig. 19, TPE-2MO2NT showed two absorption peaks in the THF solution. The peak around 343 nm was assigned to the π-π* transition, while the broad absorption band ranging from 377 to 500 nm was ascribed to the intramolecular charge transfer absorption. Meanwhile, a weak emission at 615 nm was detected for TPE-2MO2NT solution with a quantum yield (QY) of 2.2%. When a poor solvent of water was added to the THF solution, the emission intensity gradually increased due to the formation of aggregate. The highest PL intensity was recorded at aggregate state (90% water fraction) with a high QY of 42.5%, indicating its typical AIE property (Fig. 2a and Supplementary Fig. 20). Moreover, the fluorescent spectra of TPE-2MO2NT in different solvents showed polarity depended on fluorescence change, suggesting the existence of the TICT effect (Fig. 2b). From toluene to acetone, the emission maximum peaks changed from 580 to 692 nm, indicating a broad color change range of TPE-2MO2NT upon microenvironment change.

### Multiple fluorescence Colors with time-dependent
Interestingly, the emission of TPE-2MO2NT aggregates in THF/$H_2O$ ($f_w = 90\%$) mixtures obviously changed from orange to blue upon UV irradiation, demonstrating the photofluorochromic property of TPE-2MO2NT. The intriguing light response behavior of TPE-2MO2NT inspired us to further investigate the photochemical process in detail. The UV-vis absorption measurement indicated that the absorption of TPE-2MO2NT considerably blue-shifted and weakened upon UV irradiation (Supplementary Fig. 21). In particular, the two original absorption peaks around 345 and 392 nm progressively disappeared

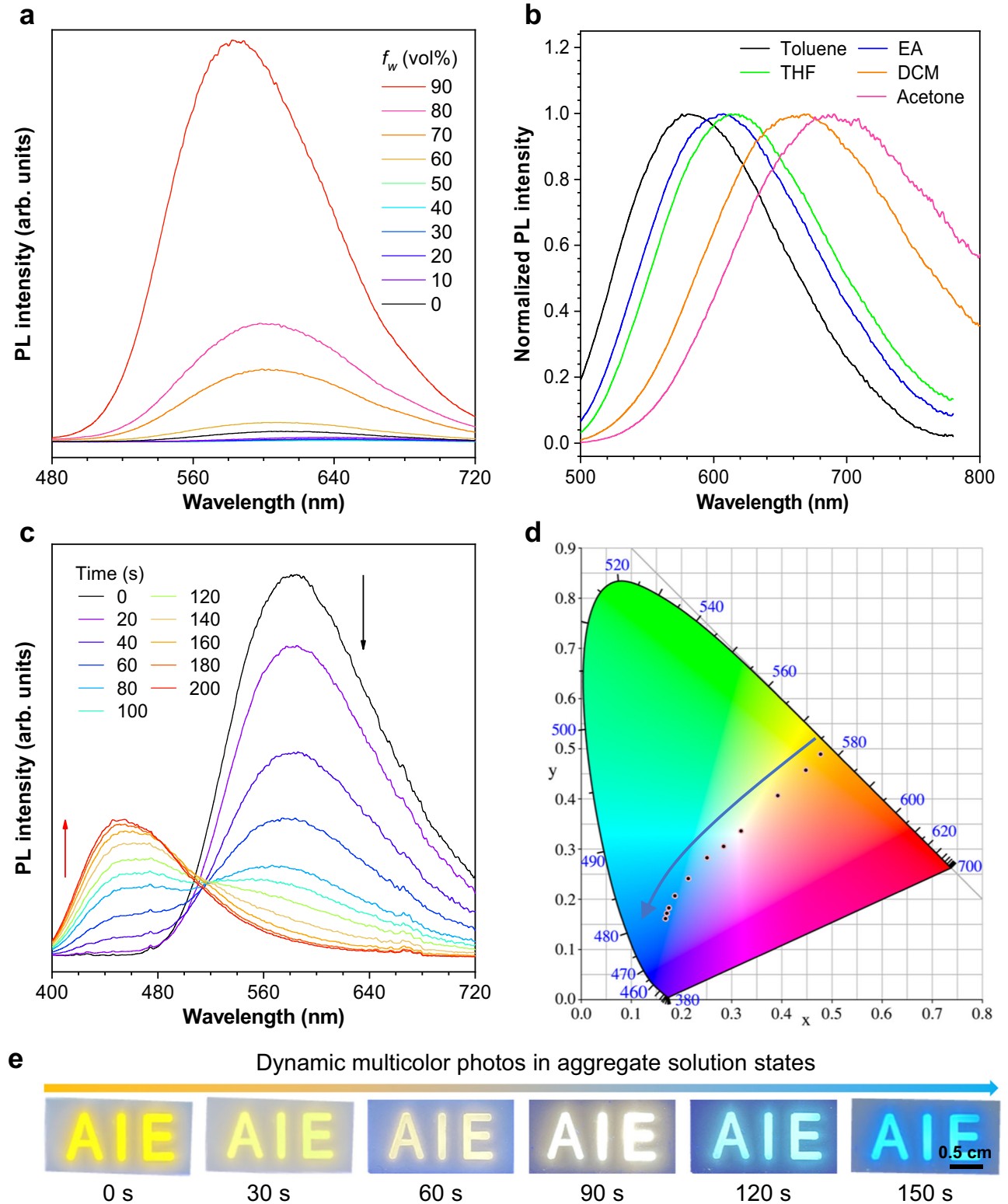

**Fig. 2 | Photophysical Properties of TPE-2MO2NT. a** PL spectra of TPE-2MO2NT in THF/H₂O mixtures with different water fractions. (**b**) Normalized PL spectra of TPE-2MO2NT in different polar solutions including toluene, ethyl acetate, tetra-hydrofuran, dichloromethane, and acetone. **c** PL spectra and **d** CIE 1931 chromaticity diagram of TPE-2MO2NT in THF/H₂O ($f_w$ = 90%) aggregate mixtures with different times upon 365 nm UV irradiation. **e** Fluorescent images of multicolor information AIE made from TPE-2MO2NT aggregate mixtures irradiated by UV light and all images share the same scale bar of 0.5 cm. Molecular concentration: $10^{-5}$ M; $\lambda_{ex}$ = 380 nm.

with irradiation time while two absorption peaks around 282 and 349 nm appeared. Meanwhile, the fluorescence intensity at 582 nm decreased and a peak at 452 nm emerged, showing an obvious hypsochromic shift of 130 nm (Fig. 2c). As presented in the CIE chromaticity diagram (Fig. 2d), the emission color exhibits successive change from orange (0.47, 0.49) to blue (0.16, 0.16), including a white light emission (0.32, 0.33). Thus, dynamic photofluorochromic behavior with high-contrast and multiple emission colors was realized in the TPE-2MO2NT photoconversion process. In contrast to other photoresponsive materials with only binary signals, the TPE-2MO2NT photoconverted system showed tunable emission signals ranging from orange to blue, including white-light emission. Encouraged by the unusual dynamic multicolor photofluorochromic property, the applications of TPE-2MO2NT aggregates for information storage and encryption were explored. As shown in Supplementary Fig. 22a, TPE-2MO2NT aggregates were filled in the AIE pattern, displaying an orange fluorescence color under UV light. The dynamic change presents six colors upon UV irradiation with time, in which only one of the AIE emission colors is the true information (Fig. 2e). Furthermore, the multicolor change could be obtained in four states using a mask to control the irradiation time of each pattern area, which realizes advanced anti-counterfeiting (Supplementary Fig. 22b, c).

Further, the photofluorochromic property of TPE-2MO2NT in THF solution was investigated by UV-vis absorption and PL emission measurements. As shown in Supplementary Fig. 23, the spectra did not change significantly under light irradiation, and no peak appeared. According to previous studies[34], the solvent has a great influence on the cleavage photoreaction, it was inferred that THF as a nonpolar solvent was unfavorable to the reaction process. In addition, white (400–700 nm) light and 405 nm light were employed to study the

photoresponsive process. Fluorescence spectra showed no emission peak appeared under the white light irradiation. The spectra showed obvious change with peak emerging upon 405 nm light irradiation, but the process is slower than that under 365 nm light irradiation (Supplementary Fig. 24). Therefore, to carry out the photoreaction efficiently, the 365 nm light was used to study the photofluorochromic process in aggregate and film states.

## Mechanism of catalyst-free oxidative cleavage photoreaction

To better understand the photoreaction process, dynamic tracing of $^1$H NMR spectroscopy of TPE-2MO2NT in CDCl$_3$ solution was performed with different UV irradiation times (Fig. 3a). With the increase of irradiation times, the peaks of H$_d$, H$_e$, H$_f$, and H$_h$ gradually decreased, while signals of H$_{b'}$, H$_{e'}$, H$_{d'}$, H$_{f'}$, and H$_{h'}$ appeared. Through separation and purification of the photoreaction products, two pure compounds named BP-2NT and BP-2MO were obtained whose chemical structures were well confirmed by NMR spectra, HRMS, and single crystal data (Fig. 3b, Supplementary Figs. 25–30, and Supplementary Tables 1, 2). The maximum photoconversions from TPE-2MO2NT to BP-2NT and BP-2MO were calculated from the $^1$H NMR spectra and reached as high as 86.7% and 80.2% after 50 h, exhibiting the efficient photoconversion process. (Supplementary Fig. 31) The resonance signals of H$_{b'}$ and H$_{d'}$ were assigned to the BP-NT proton. The peaks of H$_{e'}$, H$_{f'}$, and H$_{h'}$ were attributed to the signals of BP-2MO protons. In comparison with the methyl group signal of TPE-2MO2NT (H$_h$), the methyl group signal of BP-2MO (H$_{h'}$) exhibited a downshift due to the generation of electron-withdrawing carbonyl group. It turned out that the UV light-initiated oxidative cleavage reaction occurred in TPE-2MO2NT at ambient air without any catalyst, and two carbonyl compounds were obtained. Through analysis of

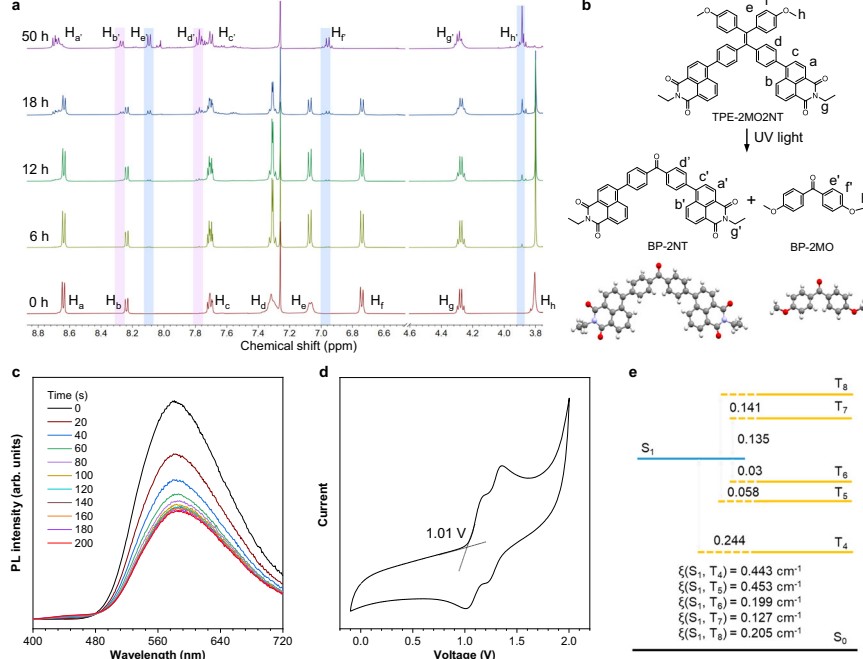

Fig. 3 | The process of photocleavage reaction was studied by experiment and theoretical calculation. a Change of $^1$H NMR spectra of TPE-2MO2NT (10 mM) in CDCl$_3$ solution under at different irradiation times. Two pink shadings indicate the characteristic peaks of the BP-2NT gradually increase over time. Four blue shadings indicate the characteristic peaks of the BP-2MO gradually increase over time. H$_a$-H$_h$ are protons of TPE-2MO2NT. H$_{e'}$, H$_{f'}$, and H$_{h'}$ are protons of BP-2MO. H$_{a'}$, H$_{b'}$, H$_{c'}$ H$_{d'}$, and H$_{g'}$ are protons of BP-2NT. b The photoinduced catalyst-free oxidative cleavage reaction route of the TPE-2MO2NT and the single crystal structure of the photoreaction products BP-2NT and BP-2MO. a-h represents the hydrogen in the position

specified in the formula of TPE-2MO2NT. a', b', c', d', and g' represent the hydrogen in the position specified in the formula of BP-2NT. e', f', and h' represent the hydrogen in the position specified in the formula of BP-2MO. c PL spectra of TPE-2MO2NT solution with triethylenediamine (scavenger for $^1$O$_2$) and benzoquinone (scavenger for O$_2^{·-}$). TPE-2MO2NT solution: TPE-2MO2NT (10 μM) in THF/H$_2$O ($f_w$ = 90%) mixtures. $\lambda_{ex}$ = 380 nm. d Cyclic voltammogram of TPE-2MO2NT in CH$_2$Cl$_2$ solution and showing the onset oxidation voltage as 1.01 V. e Energy level diagrams and SOC coefficients (ξ) for TPE-2MO2NT. S$_0$-S$_1$ are the energy levels of singlet states. T$_4$-T$_8$ are the energy levels of triplet states.

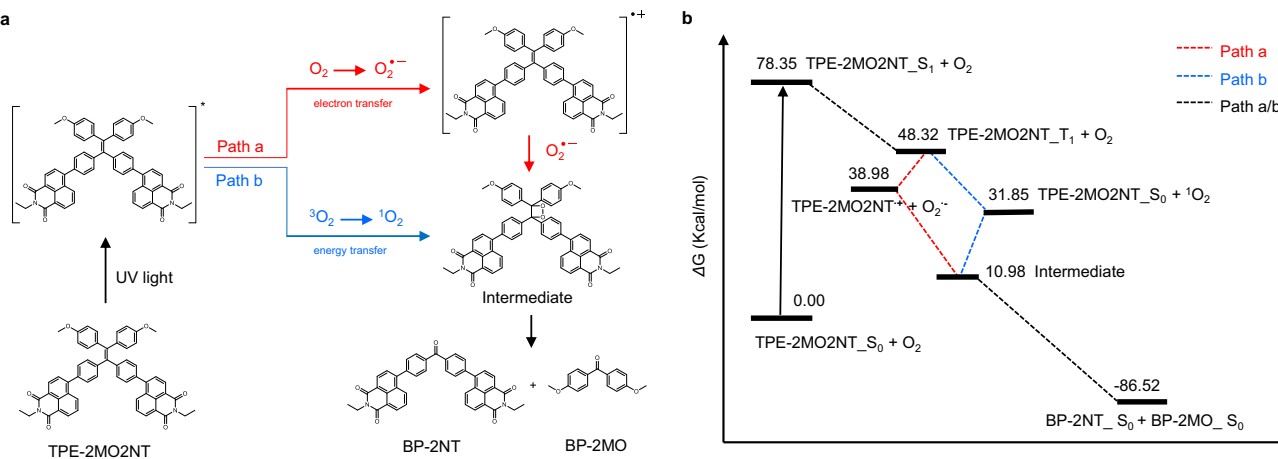

**Fig. 4 | Mechanism of catalyst-free photocleavage reaction. a** Proposed reaction mechanism and routes for the oxidative cleavage of TPE-2MO2NT. **b** Energy profiles calculated for the reaction Path **a** and Path **b**. All the numbers given were relative Gibbs free energy in kcal/mol with respect to the reference point.

photophysical data of the BP-2NT and BP-2MO (Supplementary Fig. 32), we found that BP-2NT had blue light emission with an emission peak of 451 nm in THF/H$_2$O ($f_w$ = 90%) mixtures, while BP-2MO basically was non-emissive. It could be concluded that BP-2NT is the main source of blue emission in the photofluorochromic process (Supplementary Table 3). In addition, the emission intensity of BP-2NT in aggregate state is stronger than that in pure THF solution, which demonstrates its AIE behavior (Supplementary Fig. 33). With the gradual conversion of the orange-emitting TPE-2MO2NT compound to the blue fluorophore BP-2NT, it was concluded that the multicolor photofluorochromic phenomenon is caused by the synergistic effect of TPE-2MO2NT/BP-2NT system. Compared to the common cleavage groups of o-nitrobenzene and coumarin[18,53], TPE-2MO2NT/BP-2NT as AIEgens have strong luminescence intensity and high contrast, which could allow for clear tracking and indication of the process change.

To better understand the photoreaction pathway, control experiments without UV light irradiation or in a nitrogen atmosphere were implemented which showed no apparent change (Supplementary Fig. 34). These results indicated that both light irradiation and O$_2$ are necessary to boost the oxidative cleavage reaction. It's worth noting that the reaction process can be easily traced by changes in fluorescence spectra since the generated products emit blue light and the enlarged fluorescence contrast. Further mechanism investigation indicated that the photoreaction was inhibited when benzoquinone (scavenger for superoxide radical anion O$_2^{\bullet-}$) or triethylenediamine (scavenger for singlet oxygen $^1$O$_2$) was added to the reaction system. However, the addition of tert-butyl alcohol (scavenger for •OH) did not affect the reaction (Supplementary Fig. 35). When both benzoquinone and triethylenediamine were added simultaneously into the reaction system, the blue emission peak that emerged as an indicator of the photocleavage was almost unchanged, displaying that the photoreaction process was obviously inhibited (Fig. 3c). In addition, light on-off experiments revealed that product formation was inhibited without light irradiation, thus excluding the possibility of radical chain pathway (Supplementary Fig. 36). The above experiments strongly indicated that both O$_2^{\bullet-}$ and $^1$O$_2$ are involved in the reaction process.

To explore the original source of the O$_2^{\bullet-}$ and $^1$O$_2$, 2′,7′-dichlorodihydrofluo-rescein (DCFH) as an indicator was used to test whether TPE-2MO2NT could generate ROS species under light irradiation. The fluorescence intensity of DCFH was rapidly enhanced in the presence of TPE-2MO2NT along with continuous irradiation, showing high ROS generation ability (Supplementary Fig. 37a, b). Then, dihydrorhodamine 123 (DHR123) as a common O$_2^{\bullet-}$ generation indicator

and 9,10-anthracenediyl-bis(methylene) dimalonic acid (ABDA) as $^1$O$_2$ indicator were used to test O$_2^{\bullet-}$ and $^1$O$_2$ generation ability. The increase in emission intensity (Supplementary Fig. 37c, d) and decrease in absorption (Supplementary Fig. 37e, f), respectively, indicated that TPE-2MO2NT can efficiently produce O$_2^{\bullet-}$ and $^1$O$_2$ generation. In addition, cyclic voltammetry (CV) measurement showed the onset oxidation voltage ($E_{onset}$) of TPE-2MO2NT was 1.01 V, which implies the compound could be easily oxidized (Fig. 3d). According to the above results, it was concluded that the TPE-2MO2NT could facilely generate ROS species (O$_2^{\bullet-}$ and $^1$O$_2$), which served as oxidants to realize the oxidative cleavage of the C=C bond of TPE-2MO2NT at ambient air without any catalyst. To gain more insight into this photoreaction, the theoretical calculation was implemented to get a deeper interpretation of the photoreaction. The atomic coordinates of the optimized computational compound are provided in Supplementary Data 1. As displayed in Supplementary Fig. 38, the HOMO of TPE-2MO2NT was mainly localized on the double bond, and the LUMO were primarily centered on naphthalimide groups. The obvious separation of HOMO-LUMO was beneficial to decrease the energy gap between the singlet and triplet states to increase the intersystem crossing efficiency. The calculation indicated that the energy gap between S$_1$ and T$_6$ states was as small as 0.03 eV (Fig. 3e). The fluorescence and phosphorescence spectroscopy in 77 K also showed a small energy difference between S$_1$ and T$_n$ value (Supplementary Fig. 39). The spin-orbit coupling (SOC) values between S$_1$ and T$_n$ were also provided in Fig. 3e and Supplementary Table 4. The small $\Delta E_{ST}$ value and relatively strong SOC constants benefited the intersystem crossing (ISC) process from S$_1$ to T$_n$ and facilitated the generation of triplet exciton[54]. Since the transition from T$_1$ to S$_0$ was spin-forbidden, the triplet exciton lifetime of TPE-2MO2NT was long enough to permit the sensitization of $^3$O$_2$ to produce O$_2^{\bullet-}$ (via electron transfer process) or $^1$O$_2$ (via an energy transfer process). On the other hand, since the HOMO electron clouds of TPE-2MO2NT were mainly localized on the double bond, the double bond part is most likely to react with O$_2^{\bullet-}$ and $^1$O$_2$ which have strong electron-withdrawing ability and to be cleaved, demonstrating the high selectivity and specificity of the oxidative cleavage photoreaction.

Based on literature reports[34] and comprehensive experimental studies, a proposed mechanism is illustrated in Fig. 4a. The oxidative cleavage photoreaction process involves two pathways, named Path a and Path b. In Path a, upon photoexcitation, a singlet-excited state was formed and rapidly transformed into triplet state via ISC owing to a small $\Delta E_{ST}$ value and high SOCs. The excited TPE-2MO2NT could easily transfer electrons to O$_2$ through the electron transfer process to generate O$_2^{\bullet-}$. Then, O$_2^{\bullet-}$ as a strong oxidant attacked the C=C bond to generate a dioxetane intermediate that underwent [2 + 2]

cycloaddition. Finally, the dioxetane was cleaved to produce two carbonyl products BP-2NT and BP-2MO. To demonstrate the thermodynamic feasibility of reaction pathways, their Gibbs free energies were calculated. The Gibbs free energies of BP-2NT and BP-2MO are lower than that of TPE-2MO2NT and $O_2$, indicating that Path a is allowed (Fig. 4b). For Path b, $^1O_2$ was generated via the energy transfer between triplet TPE-2MO2NT and $^3O_2$. Then, the resultant $^1O_2$ reacted with TPE-2MO2NT to produce dioxetane intermediate. After intermediate cleavage, two carbonyl products BP-2NT and BP-2MO were also finally generated. The energy profiles of Path b shown in Fig. 4b indicated that Path b is also allowed. In general, the excited TPE-2MO2NT by light acted as a triplet sensitizer to produce highly reactive $O_2^{\cdot-}$ and $^1O_2$, which subsequently oxidized the C=C bond of the substrate and cleavage to carbonyl compounds. In contrast to other catalytic-free oxidative cleavage systems focusing on the organic synthesis aspect[34], this study systematically investigated the photoreaction mechanism and applied the photoreaction to construct the TPE-2MO2NT photocleavage skeleton (Supplementary Fig. 40).

## The design strategy of TPE-2MO2NT molecular skeleton

Whether the photocleavage of TPE-2MO2NT also applicable to other tetraphenylethene derivatives? For further deep understanding of oxidative cleave photoreaction, two other compounds, TPE-4MO and TPE-4NT, were synthesized and well characterized by NMR spectroscopy (Supplementary Figs. 41–46). Compared with TPE-2MO2NT containing two electron-donating methoxy groups and two electron-withdrawing naphthalimide units, TPE-4MO is modified with four methoxy groups and TPE-4NT has four naphthalimide substituents (Fig. 5). TPE-4MO exhibited typical AIE phenomenon showing a bright blue-green emission in THF/$H_2O$ ($f_w = 90\%$) mixtures (Supplementary Fig. 47). However, the PL spectra of TPE-4MO tended to blue shift and decreased in intensity with the increase of 365 nm light irradiation time. The change also can be vividly seen from photos of TPE-4MO system before and after 200 s 365 nm light irradiation (Fig. 5). The absorption spectra of TPE-4MO also changed, and two weak absorption peaks at around 260 and 329 nm decreased and absorption at around 255 nm appeared. $^1H$ NMR spectra with different irradiation times were carried out to find out the change process (Supplementary Fig. 48). The aromatic protons with chemical shifts of 6.67 and 6.83 ppm disappeared, while signals at 6.82, 7.03, 7.18, 7.28, and 8.23 ppm appeared. According to spectra analysis and literature results[55], it turned out that TPE-4MO underwent photocyclization reaction and transformed into 9,10-diphenylphenanthrene derivative. For the TPE-4NT, the emission intensity in THF/water mixtures ($f_w = 90\%$) obviously enhanced compared with its pure THF solution, exhibiting typical AIE features (Supplementary Fig. 49). While the absorption and emission spectra of TPE-4NT showed no apparent change over time under light irradiation (Fig. 5). The dynamic $^1H$ NMR spectra also display no distinct change within 20 hours of irradiation time (Supplementary Fig. 50).

To further understand the different photoreactions, theoretical calculations, ROS evaluation, and CV measurement of TPE-4MO and TPE-4NT were conducted. The data are summarized in Table 1.

For TPE-4MO, the large energy gap between $S_1$ and $T_n$ and small SOC coefficients would weaken the ISC efficiency, which is not conducive to the generation of triplet exciton and ROS (Supplementary Fig. 51 and Supplementary Table 5). Indeed, ROS generation experiment indicated that only a slight change was shown for the fluorescence signal of the DCFH indicator, suggesting TPE-4MO produced almost no ROS (Supplementary Fig. 52a, b). Although the CV test showed that TPE-4MO had a relatively low initial oxidation potential (0.92 V), the oxidative cleavage photoreaction did not occur due to the absence of ROS generation capability in the system (Supplementary Fig. 52c). For TPE-4NT, its small $\Delta E_{ST}$ and high SOC coefficient would enhance the ISC channel, which is beneficial to the generation of triplet excitons (Supplementary Fig. 53 and Supplementary Table 6). Further ROS generation measurement indicated that TPE-4NT indeed could produce ROS, while its ability was weaker than TPE-2MO2NT (Supplementary Fig. 54a, b). However, TPE-4NT has a higher onset oxidation potential at 2.06 V according to the CV test (Supplementary Fig. 54c), which makes TPE-4NT much more difficult to oxidize due to its high oxidation potential. According to the above results, we speculated that the introduction of the electron-donating methoxy group can reduce oxidation potential and make the compound susceptible to oxidation; On the other hand, the electron-donating and electron-withdrawing structure could endow the compound with the ability to produce ROS as reactive oxidants. These two reasons promote TPE-2MO2NT to undergo oxidative cleavage with light irradiation, accompanied by noticeable fluorescence change.

Further, a series of other different electron-donating and withdrawing groups were decorated on tetraphenylethylene skeleton and studied the changes of these compounds under light irradiation. The chemical structures of five compounds were shown in Supplementary Fig. 55 and fully characterized spectroscopically (Supplementary Figs. 56–70). By comparing the $^1H$ NMR spectra before and after illumination (Supplementary Figs. 71–75), the two compounds (TPE-2M2NT and TPE-2MO2CN) could undergo the oxidative cleavage reaction. However, the reaction efficiencies of these two compounds were very low compared with TPE-2MO2NT. The other two compounds (TPE-2M2Ph and TPE-2MO2AB) mainly experienced photocyclization reaction, and the TPE-2M2NB had no obvious peak change of $^1H$ NMR spectra under light. The results showed that the photocleavage, photocyclization, and photostabilization of TPE derivatives can be regulated by different substituents, which also could provide useful information for the photoresponsive design of other TPE compounds. Through the theoretical calculation and ROS test analysis (Supplementary Figs. 76–80 and Supplementary Table 7), we speculated that although the compounds had similar HOMO energy levels and ROS generation efficiency, there was no effective oxidative cleavage reaction which may be related to the reaction energy barrier, the stability of the intermediate and other factors. Additionally, the UV-Vis, PL spectra, and quantum yield test were conducted to analyze the photofluorochromism (Supplementary Figs. 81-85). The summarized data are shown in Supplementary Table 7. Compared with the change of emission peak before and after illumination, TPE-2MO2NT had the largest wavelength shift. And the emission intensity of the five compared compounds after light irradiation was weak and the fluorescence quantum yield was relatively low, which is not conducive to practical application. In terms of reaction efficiency and photochromic performance, TPE-2MO2NT displayed high photooxidative cleavage reaction efficiency and excellent tunable multicolor emission, indicating the design of TPE-2MO2NT skeleton is distinctive and significant.

To show the versatility of the TPE-2MO2NT skeleton, two derivatives of TPE-2MO2NT, TPE-MN-C5 and TPE-MN-Br, were prepared with different alkyl chains (Supplementary Figs. 10, 11, Supplementary Figs. 86–91). According to the absorption, PL spectra, and CIE diagrams (Supplementary Figs. 92, 93), TPE-MN-C5 and TPE-MN-Br also

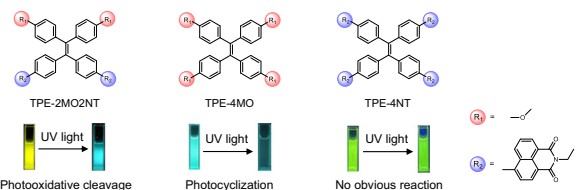

**Fig. 5 | Comparison of the photoreactions of three compounds.** Chemical structures and fluorescent photographs of TPE-2MO2NT, TPE-4MO, and TPE-4NT before and after UV light irradiation.

**Table 1 | Summary data of the three compounds with different photoreactions**

| Compound | HOMO[a] (eV) | LUMO[a] (eV) | $\Delta E_{st}$[b] (eV) | $\xi(S_1, T_n)$ (cm$^{-1}$)[c] | ROS efficiency[d] | $E_{onset}$[e] (V) |
|---|---|---|---|---|---|---|
| TPE-2MO2NT | −6.62 | −1.67 | 0.033 | 0.453 | 203.63 | 1.01 |
| TPE-4MO | −6.31 | −1.78 | −0.124 | 0.254 | 4.40 | 0.92 |
| TPE-4NT | −7.12 | −0.26 | 0.005 | 0.351 | 89.42 | 2.06 |

[a]: HOMO and LUMO levels are calculated by DFT. [b]: smallest $S_1$ and $T_n$ energy gap. [c]: maximum values of SOC coefficients (ξ) between specific $S_1$ and $T_n$. [d]: Relative PL intensity of DCFH with the compound before and after irradiation, respectively. [e]: onset oxidation potential from CV curves.

showed typical AIE effect and displayed controllable multicolor photofluorochromic phenomenon with UV light irradiation. To expand the application of TPE-2MO2NT derivatives, the TPE-MN-TA compound was demonstrated and modified by triazole substituents (Supplementary Fig. 12, Supplementary Figs. 94–96). TPE-MN-TA also displayed multicolor photofluorochromic behaviors (Supplementary Fig. 97) and had excellent antibacterial effects. *S. aureus* and *E. coli*, as the representatives of Gram-positive (G+) and Gram-negative (G-) bacteria, were used to study the bactericidal effects. As shown in Supplementary Fig. 98, TPE-MN-TA effectively inhibited not only G- bacteria but also G+ bacteria at a relatively low concentration, indicating its potential in bioimaging and therapy. These results indicate that the TPE-2MO2NT molecular skeleton is easy to modify. The photocleavable luminescent skeleton and its cleavage photoreaction could provide a starting point for future photoresponsive application possibilities, like photoinduced micropatterns, controlled drug delivery, and photodegradable materials. To enrich the scope and applications of TPE-2MO2NT photocleavable skeleton photodegradation frameworks, it is worth studying and continuing to focus on the introduction of appropriate substituents to modify TPE-2MO2NT in the future.

**Dynamic Pattern and Time-gated 4D Codes**

As a proof of concept, we further explored TPE-2MO2NT potential for developing advanced functional systems encouraged by the multiple fluorescence changes with irradiation time-dependent. Three kinds of TPE-2MO2NT-based films were prepared, including pure TPE-2MO2NT film, doped TPE-2MO2NT/PMMA film, and doped TPE-2MO2NT/PEG film. We speculated that the rigidity of different doping polymers would endow TPE-2MO2NT with different molecular microenvironments to achieve the tuning of emission behaviours of the films and precisely control the speed of photoreaction. As expected, pure TPE-2MO2NT film emitted an orange-yellow color at 568 nm, while it showed green and orange color at 516 and 587 nm in the PMMA matrix and PEG matrix, respectively (Supplementary Fig. 99). The distinct emission colors in the three films were attributed to the TICT effects of TPE-2MO2NT[56]. In general, a rigid polymer matrix will restrict the excited state molecular motion and thus inhibit the access to the long wavelength but dark TICT state, and thus usually afford relatively strong and blue-shifted emission. Furthermore, according to emission spectra and CIE diagrams, the three films display different response times under UV irradiation (Supplementary Fig. 100). Compared with pure TPE-2MO2NT film, TPE-2MO2NT/PMMA film showed little color change in a longer time but TPE-2MO2NT/PEG film exhibited an obvious color change in a shorter time. As PMMA is a rigid polymer and PEG is a soft polymer, a relatively large free volume between PEG polymer chains benefits TPE-2MO2NT to undergo photoreaction, while the photochromic process in the rigid PMMA polymer environment with more constraint needs more response time. Due to the photoproduct BP-2NT in different matrices could affect the degree of aggregation, pure BP-2NT film, doped BP-2NT /PMMA film, and doped BP-2NT /PEG film showed different emission colors after light irradiation (Supplementary Fig. 101–103). It means that the different luminous colors of the pattern would also be obtained after the light irradiation. Thus, TPE-2MO2NT in film states showed multiple controllability based on the influence of the polymer matrix in the photoreaction process.

Further, we explored the potential of TPE-2MO2NT for dynamic fluorescence image application encouraged by the controllable photofluorochromic phenomenon in the film state. As shown in Fig. 6, a tower pattern was placed on the glass sheet and 1 mg/mL CHCl$_3$ solution of TPE-2MO2NT was sprayed in the tower area. A tower image was fabricated with orang-yellow emission color. With increasing UV light irradiation time, the florescent color of the tower image exhibited obvious change within 270 s and finally turned into a blue-green color, demonstrating a dynamic multicolor image was achieved. Through the above procedure, a four-leaf clover image was prepared by spray coating CHCl$_3$ solution of TPE-2MO2NT/PMMA with a weight ratio of 1:100 on a glass sheet. The initial green color image underwent oxidative cleavage photoreaction and varied with light irradiation. Due to the rigid PMMA polymer matrix, the florescent color of four-leaf clover image changed slowly and a blue-green color image was achieved in 360 s. Similarly, an orange florescent color clover image was manufactured through the spin coating method with CHCl$_3$ solution of TPE-2MO2NT/PEG with a weight ratio of 1:100. Compared with the other two films, the photoreaction process of this film in soft PEG polymer matrix changed faster, finally getting a blue clover image within 180 s.

Inspired by the above dynamic fluorescence imaging, 4D anti-counterfeit code with dynamic information storage and encryption capability was explored. In comparison with 2D code, 4D code increased the dimensions of time and space (e.g. dynamic color change with time), which greatly improved the density of information storage and cracking difficulty of 4D anti-counterfeit code[57–59]. As shown in Fig. 7a, the films could be fabricated by spin coating with CHCl$_3$ solution of TPE-2MO2NT, TPE-2MO2NT/PMMA, and TPE-2MO2NT/PEG, respectively. Then, three kinds of film were combined into a square pattern composed of sixteen independent films. The original square pattern showing three different emission colors was defined as code 1. Due to the different photoreaction rates of the three kinds of films, the code could dynamically transform into code 2, code 3…code N with the extension of the UV light irradiation time. The 4D codes information could be read by a smartphone upon UV irradiation.

Based on the above conceptual design, we fabricated time-gated 4D codes using a combination of TPE-2MO2NT film, TPE-2MO2NT/PMMA film, and TPE-2MO2NT/PEG film (Fig. 7b). The original multicolor fluorescent pattern constituted code 1 and a series of dynamic codes were achieved with increasing irradiation time. The areas of TPE-2MO2NT/PEG film showed noticeable color change after 1 min irradiation due to its fast photocleavage rate. Accordingly, code 1 transformed into code 2. After another 1 min of exposure, the areas of TPE-2MO2NT/PEG film turned mostly blue and the areas of TPE-2MO2NT film also showed some discoloration, forming code 3. With a further extension of 1 min irradiation time, the areas of TPE-2MO2NT film almost showed green color, and code 4 was generated. Finally, code 5 with multicolor emission was obtained within 4 min irradiation. Notably, time-gated 4D codes were successfully achieved with the dynamical transformation of code 1 to code 5 with increasing light irradiation time. By defining code 2 and code 5 as the true information and the other three codes as wrong information, the true information could be identified at only specified irradiation times (2 min and 4 min) and other occurred information is wrong during the rest of the time course, showing high-level security information storage and

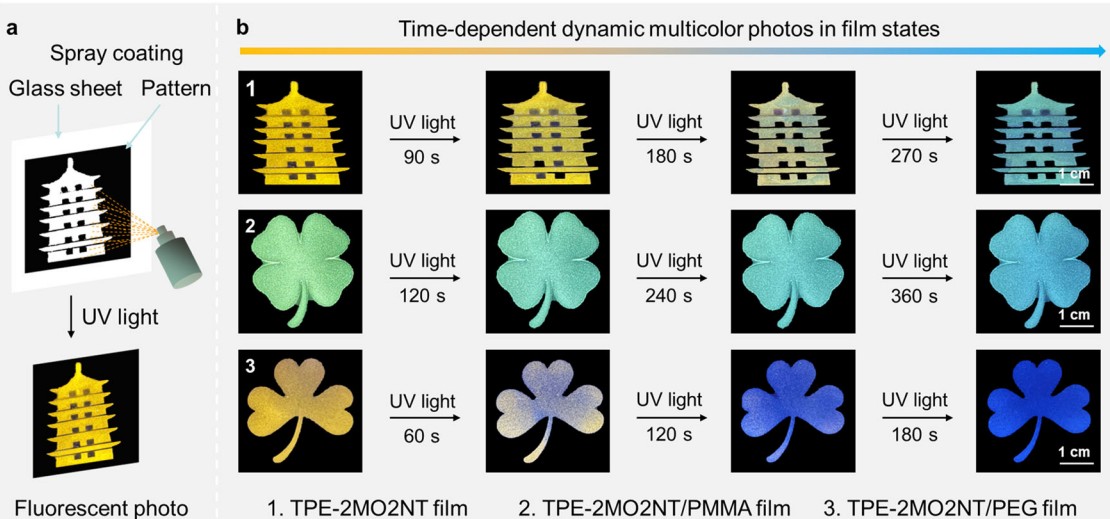

**Fig. 6 | Dynamic multicolor patterning with time-dependent. a** Fabrication process for fluorescent photos using photopatterning method. **b** Fluorescent photographs of the obtained pattern on a glass sheet from three different solutions (CHCl₃ solution of TPE-2MO2NT, TPE-2MO2NT/PMMA, and TPE-2MO2NT/PEG) and their responsive behaviors along with time under UV irradiation, respectively. All images share the same scale bar of 1 cm.

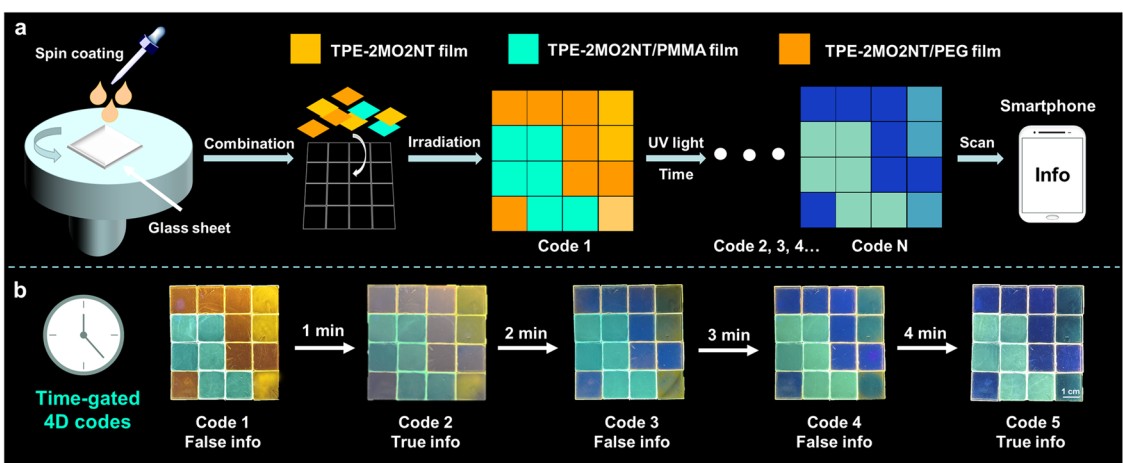

**Fig. 7 | Time-gated 4D codes for advanced encryption. a** Schematic diagrams of the fabrication and scan steps for information codes. **b** Fluorescent photographs of the time-gated 4D codes under UV light are presented. All images share the same scale bar of 1 cm.

protection capability. Since the photoreaction process can be controlled by parameters, such as light intensity, emission wavelength, and exposure distance. Therefore, the time for acquiring information by irradiation of the 4D code can be changed. For example, by increasing the light intensity to shorten the time for obtaining information. It can be judged that with the adjustment of the parameters, the 4D information anti-counterfeiting system can be improved.

## Discussion

In this work, the multiple photofluorochromism of TPE-2MO2NT converted system which could undergo catalyst-free photooxidative cleavage reaction under UV irradiation was developed. The photo-induced oxidative cleavage reaction of TPE-2MO2NT was highly efficient and specific. As the orange emitting TPE-2MO2NT gradually cleavage to the blue fluorophore BP-2NT, the system endowed high contrast and tunable multicolor emission. Through systematical experiments and theoretical analysis, this photofluorochromic process mechanism was well revealed, in which TPE-2MO2NT served as a triplet sensitizer to produce highly reactive $O_2^{\cdot-}$ and $^1O_2$ under light irradiation and then cleave the C=C bond of the substrate to produce two

BP-2NT and BP-2MO compounds. Through comprehensive analysis of TPE-2MO2NT and the other compared molecules, it is demonstrated that the TPE-2MO2NT molecular skeleton is suited for exploitation. Furthermore, TPE-2MO2NT showed multicolor microenvironmental controllability in the polymer matrix. The time-dependent dynamic images and 4D information codes application were achieved, in which true information can be acquired at only a specified time, showing advanced anticounterfeiting function. Our work not only comprehensively studied the mechanism of catalyst-free oxidative cleavage reaction, but also enriched the library of photocleavage molecular skeleton and photoresponsive mechanism. We believe these discoveries could provide dynamic controllable fluorescence materials but also enable more exciting photoresponsive applications like surface patterning, controlled drug delivery, and polymer degradation with the TPE-2MO2NT molecular skeleton.

## Methods

### Fabrication procedure of time-gated 4D codes

TPE-2MO2NT and polymer stock solutions were prepared by dissolving 1 mg of TPE-2MO2NT in 1 mL CHCl₃ and dissolving 50 mg of

the polymer sample in 1 mL of $CHCl_3$, respectively. Afterward, 0.1 mL of TPE-2MO2NT solution was mixed with 0.2 mL of polymer solution under sonication for about 0.5 h to produce a homogeneous solution doped with the 1.0 wt% TPE-2MO2NT. The films of TPE-2MO2NT doped polymers were fabricated by spin coating the mixed solutions of TPE-2MO2NT and polymer blend onto glass sheets (1.5 cm × 1.5 cm). The rotational speed for spin coating is 500 rounds/min for 5 s and then 1500 rounds/min for 30 s. The TPE-2MO2NT films were fabricated by spin coating the TPE-2MO2NT stock solutions onto glass sheets (1.5 cm × 1.5 cm) at 500 rounds/min for 5 s and then 1500 rounds/min for 30 s. The obtained films were made in a specific combination to obtain the 4D codes and then irradiated to get different information.

## Data availability

The authors declare that all the data supporting the findings of this manuscript are available within the manuscript and Supplementary Information files and available from the corresponding authors upon request. The Cartesian coordinates of optimized molecular geometry are provided in Supplementary Data 1. The X-ray crystallographic coordinates for structures reported in this study have been deposited at the Cambridge Crystallographic Data Centre (CCDC), under deposition numbers 2256097-2256098. These data can be obtained free of charge from The Cambridge Crystallographic Data Centre via www.ccdc.cam.ac.uk/data_request/cif.

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

## Acknowledgements

This work was financially supported by the National Natural Science Foundation of China NSFC (52003228, 52273197, and 22271197), Shenzhen Key Laboratory of Functional Aggregate Materials (ZDSYS20211021111400001), the Science and Technology Plan of Shenzhen (JCYJ2021324134613038, KQTD20210811090142053, JSGG20220606141800001, GJHZ20210705141810031, RCYX20221008092924059, KQTD20210811090142053, JCYJ20220531102601003, JCYJ20220818103007014), the Guangdong Basic and Applied Basic Research Foundation (2023A1515011578, 2022A1515110950), The authors also acknowledge the Instrumental Analysis Center of Shenzhen University.

## Author contributions

L.L., Z.Z., and B.Z.T. conceived and designed the experiments. L.L. performed the experimental work. B.W. conducted the theoretical calculation. F.Z. conducted the antibacterial test. X.F. performed the single-crystal measurements. L.L., Z.Z., X.H., T.H., Z.Q., D.W., and B.Z.T. contributed to manuscript writing and editing. All authors approved the final version of the manuscript.

## Competing interests

The authors declare no competing interests.
