## [Peer Review File · Nature Communications]

Multiple photofluorochromic luminogens via catalyst-free alkene oxidative cleavage photoreaction for dynamic 4D codes encryptionREVIEWER COMMENTS

Reviewer #1 (Remarks to the Author):

Photo-responsive materials represent a kind of smart materials which aroused broad research interest in sensors, supramolecular chemistry, biomedical area etc. Exploration of new photo-responsive material system through new design principle is a challenge of this area. In this work, Tang et al reported an interesting aggregation-induced emission luminogen that underwent catalyst-free oxidative cleavage under light irradiation. The photocleavage reaction endows the system with tunable multicolor behaviors and its reaction mechanism was studied thoroughly. The mechanism proved the TPE-2MO₂NT could self-produce highly reactive oxidants to undergo oxidative cleavage reaction by light irradiation. This photocleavage skeleton is unique and also proved to be modified. The applications of dynamic patterns and 4D codes were presented, showing the potential of anti-counterfeiting applications. In general, the current work demonstrates a new strategy to afford photo-responsive optical materials and the photocleavage mechanism has been thoroughly investigated. The data could well support their conclusion and the manuscript is well presented. I think the manuscript could arouse a broad interest of readers of related area. I do have some suggestions that could further improve the quality of the manuscript. I suggest a publication in Nature Communications after the authors addressed the following questions:

1. Because TPE-2MO₂NT can undergo photocleavage reaction and is different from other common TPE derivatives, the molecular design strategy of TPE-2MO₂NT should be explained in the manuscript.
2. The authors studied the fluorescent change in aggregate and film states. To fully understand this change, it also should be provided the light irradiation behaviors in dilute solutions like in THF solution.
3. The photocleavage reaction was studied by 365 nm light irradiation. Since the compound has a certain absorption in the visible range, is it possible that photocleavage can also occur under visible light irradiation?
4. TPE-2MO₂NT has a small S₁ and T_n energy gap according to theoretical calculation. It is preferable to provide the phosphorescence behavior of TPE-2MO₂NT.
5. The excitation wavelength of the emission spectra should be provided in the figure caption.
6. In Figure 4 and Figure 5, different colors of green and blue appeared in different polymer matrices after they were illuminated. Please explain this phenomenon and supplement relevant spectral tests.
7. The fabrication of 4D information codes should be described in detail, like the concentration of solution, the speed of spin coating, and the size of the glass sheet.
8. As TPE-2MO₂NT is a new class of cleavage skeleton, the authors can compare with the previous cleavage groups to highlight the advantages of the TPE-2MO₂NT compound.

Reviewer #2 (Remarks to the Author):

The only aspect I am able to assess is the synthetic reactivity reported in terms of olefin cleavage since I

am not a material scientist.

From that perspective the work in sound and the results obtained are supportive of the proposed mechanism based on singlet oxygen generation and following olefin cleavage.

However, this is nothing noteworthy from a synthetic point of view since olefin oxidative cleavage using singlet oxygen can only be achieved on very activated systems like tetra-aryl substituted olefins.

Overall, there is not much of relevance to the synthetic community from this work, which however seems more directed to material science. The impact on this field is something that goes beyond my expertise.

Reviewer #3 (Remarks to the Author):

In their manuscript, Lu et al. describe a novel photofluorochromic system based on a highly emissive AIEgen which can be photooxidized selectively under UV radiation to form a different fluorescent species. The authors dutifully characterize the chemical and optical properties of the system and employ DFT methods not only to rationalize the mechanisms but also to explore the properties of a series of similar compounds, identifying the unique properties of TPE-2MO₂NT. Finally, they show how it is possible to use the unique features of this system and the color versatility offered by AIE to prepare time-gated photofluorochromic substrates that could find application in anticounterfeiting and cryptography.

The system presented is in itself very interesting and well characterized, as such it can be appealing to broad audiences, spanning both basic and applied fields. However, I find that, in many instances, the study is 'oversold' and the authors claim more novelty than what is actually there. As such I do not believe that the study is suitable for the standard of Nature Communications.

In particular, I believe that it is misleading to say that the compound TPE-2MO₂NT is by itself responsible for the observed photofluorochromic phenomenon as suggested in the introduction (e.g. lines 103-106, 112-114) and repeated throughout the text (e.g. L161-163). The 'continuous fluorescence change' originates from the disappearing of the AIEgen fluorescence at 580 nm and the appearance of a blue emission at 460 nm ascribable to one of the products of its oxidation. In other words, the continuous fluorescence is due to the synergic effect of two different chemical species, one slowly disappearing and one slowly forming (due to the oxidation of a compound to yield another). This is effectively an irreversible photochromic effect, and, as such, these two contributions should not be regarded only from the perspective of the starting compound (i.e. the photofluorochromisms of TPE-2MO₂NT) but in light of the (irreversible) chemical change (i.e. the photofluorochromism of the TPE-2MO₂NT/BP-2NT system) which yields a blue fluorophore (as they acknowledge in L196).

In this sense, the system presented here is somewhat different and should not be compared directly to photochromism phenomena in which the unimolecular nature is obvious (which the author references in ref 10-17 to discuss the state of the art of photofluorochromism) and which are, I believe, more interesting as a result of the fact that more complex colorations can be obtained by rational combination of more entities (see ref 55). The approach the authors decided to follow to describe their system leads to several inconsistencies, such as in L169 and Figure 1e in which the authors argue that the system

presents 6 colour changes while having argued before that they obtain a gradual colour shift.

The mechanism of the photooxidation is very similar to what reported in the case of ref 33. (Such mechanism should be added to Scheme 1 for a fair comparison with the state-of-the-art.) While the authors have done an impressive work to investigate the mechanism – combining DFT results, control experiments, and expanding the rationale to structurally related compounds – the latter is not new per se and only applies to a small set of compounds, preventing the generalization to design rules that might guide the synthesis of similar systems.

Finally, compared to other previous works of the authors (e.g., 10.1002/anie.202208460, 10.1038/s41467-023-38801-1, 10.1002/adfm.202213927) and those referenced in the study, I found the proposed anti-counterfeiting application rather underwhelming. While I agree with the rationale behind the time-evolving codes, the one proposed by the authors and described in Figure 5 suffers some drawbacks. For example, the readout input is the same as the stimulus that triggers the change and as such, unless the only true information will be the last code generated, some information (in Fig 5 the authors identify code 2 also as a true info) can only be read for a short period of time, after which is irreversibly lost. In addition, it takes several minutes to get to the final (true) configuration so information retrieval will be quite slow and not functional. On top of that, once such configuration is reached, as the fluorescence won't change anymore, looking through UV light will reveal the right info and in this sense the cryptographic key is only one-use.

Responses to the Comments and Suggestions of Reviewers

Response to the Comments of Reviewer 1:

Comments: Photo-responsive materials represent a kind of smart materials which aroused broad research interest in sensors, supramolecular chemistry, biomedical area etc. Exploration of new photo-responsive material system through new design principle is a challenge of this area. In this work, Tang et al reported an interesting aggregation-induced emission luminogen that underwent catalyst-free oxidative cleavage under light irradiation. The photocleavage reaction endows the system with tunable multicolor behaviors and its reaction mechanism was studied thoroughly. The mechanism proved the TPE-2MO₂NT could self-produce highly reactive oxidants to undergo oxidative cleavage reaction by light irradiation. This photocleavage skeleton is unique and also proved to be modified. The applications of dynamic patterns and 4D codes were presented, showing the potential of anti-counterfeiting applications. In general, the current work demonstrates a new strategy to afford photo-responsive optical materials and the photocleavage mechanism has been thoroughly investigated. The data could well support their conclusion and the manuscript is well presented. I think the manuscript could arouse a broad interest of readers of related area. I do have some suggestions that could further improve the quality of the manuscript. I suggest a publication in Nature Communications after the authors addressed the following questions:

Overall Reply: We greatly appreciate the positive comments from the reviewer on our work. We have carefully read each of the comments and tried our best to address them. Our point-by-point response is given below.

1. Because TPE-2MO₂NT can undergo photocleavage reaction and is different from other common TPE derivatives, the molecular design strategy of TPE-2MO₂NT should be explained in the manuscript.

Response: Thanks for the suggestions of the reviewer. The sentence “TPE-2MO₂NT integrates three functionalities into the molecular design: luminescent unit for observing fluorescence change; ROS generation unit for producing reactive oxidant; activated alkene for oxidizing cleavage (Scheme 1).” was added in the revised manuscript.

2. The authors studied the fluorescent change in aggregate and film states. To fully understand this change, it also should be provided the light irradiation behaviors in dilute solutions like in THF solution.

Response: Thanks for the reviewer’s constructive suggestions. The photofluorochromic property of TPE-2MO₂NT in THF solution was studied by the UV-vis absorption and PL emission measurement. As shown in Figure C1, the absorption and fluorescence spectra displayed no obvious change, and no new peak appeared with light irradiation.

According to the previous study that the solvents had a great influence on the cleavage photoreaction (Green Chem., 2021, 23, 3649-3655), it was inferred that THF as a nonpolar solvent was not conducive to the reaction process. Since the TPE-2MO2NT as typical AIEgen showed high emission intensity in aggregate state, we mainly studied the photofluorochromic behavior in aggregate and film states. The sentence “Further, the photofluorochromic property of TPE-2MO2NT in THF solution was investigated by UV-vis absorption and PL emission measurements. As shown in Figure S11, the spectra did not change significantly under light irradiation, and no new peaks appeared. According to previous studies³⁴, the solvent has a great influence on the cleavage photoreaction, it was inferred that THF as a nonpolar solvent was unfavorable to the reaction process.” was added in the revised manuscript and the Figure C1 was added in the revised supporting information.

Figure C1. (a) UV-vis absorption and (b) PL spectra of TPE-2MO2NT in THF solution with different times upon 365 nm UV irradiation. Molecular concentration: 10^{-5} M; $\lambda_{\text{ex}} = 380$ nm.

3. The photocleavage reaction was studied by 365 nm light irradiation. Since the compound has a certain absorption in the visible range, is it possible that photocleavage can also occur under visible light irradiation?

Response: We thank the reviewer for the valuable question. We used a 405 nm lamp and a white light (400-700 nm) lamp to illuminate the sample and characterize it by spectrum. As shown in Figure C2, fluorescence spectra showed little change without new emission peak appearing under the white light illumination. When the 405 nm lamp was utilized, there were some obvious changes in PL spectra, indicating that the photocleavage reaction can occur. However, compared with the 365 nm lamp irradiation, it was found that the speed of change was slower. We hypothesize that it may be caused by the gradual disappearance of absorption charge transfer bands during light irradiation; The other is because 365 nm light has a higher energy than 405 nm light, making higher efficiency of the photoreaction. According to this meaningful question, we would consider the next step to modify substitutions that make this photoreaction efficiently occur under longer light irradiation, such as constructing more

conjugated molecular structures. The sentence “In addition, white (400-700 nm) light and 405 nm light were employed to study the photoresponsive process. Fluorescence spectra showed no new emission peak appeared under the white light irradiation. The spectra showed obvious change with new peaks emerging upon 405 nm light irradiation, but the process is slower than that under 365 nm light irradiation (Figure S12). Therefore, to carry out the photoreaction efficiently, the 365 nm light was used to study the photofluorochromic process in aggregate and film states.” was added in the revised manuscript and Figure C2 was added in the revised supporting information.

Figure C2. PL spectra change of TPE-2MO2NT (10 μ M) in THF/H₂O ($f_w = 90\%$) aggregate mixtures with different times upon (a) 365 nm, (b) white light (400-700 nm) and (c) 405 nm light irradiation. $\lambda_{ex} = 380$ nm.

4. TPE-2MO2NT has a small S_1 and T_n energy gap according to theoretical calculation. It is preferable to provide the phosphorescence behavior of TPE-2MO2NT.

Response: Thanks for the reviewer’s suggestion. Following the advice, the phosphorescence behavior of TPE-2MO2NT state was measured and shown in Figure C3. The fluorescence and phosphorescence spectra in 77 K showed small energy gap between S_1 and T_n value, which is consistent with theoretical calculation of TPE-2MO2NT. The sentence “The fluorescence and phosphorescence spectroscopy in 77 K also showed small energy difference between S_1 and T_n value (Figure S27).” was added in our revised manuscript and Figure C3 was added in the revised supporting information.

Figure C3. Fluorescence and phosphorescence spectra of TPE-2MO2NT powder in 77 K. $\lambda_{\text{ex}} = 380 \text{ nm}$.

5. The excitation wavelength of the emission spectra should be provided in the figure caption.

Response: Thanks for the reviewer's careful review. We carefully checked the figure captions and added the excitation wavelength in the revised manuscript and supporting information.

6. In Figure 4 and Figure 5, different colors of green and blue appeared in different polymer matrices after they were illuminated. Please explain this phenomenon and supplement relevant spectral tests.

Response: Thanks for the reviewer's suggestion. Because TPE-2MO2NT was oxidized cleavage to produce another luminous substance BP-2NT after irradiation, three kinds of BP-2NT films were studied by emission spectra, including pure BP-2NT film, doped BP-2NT /PMMA film, and doped BP-2NT /PEG film. Compared to the spectra of pure BP-2NT film and BP-2NT /PMMA film, the spectrum of BP-2NT /PEG film showed an obvious blue shift (Figure C4). Speculating that the polarity of the environment may affect luminescence, the emission of BP-2NT was tested in different polarity solvents (Figure C5). However, the spectra did not show a regular red shift with increasing solvent polarity, indicating that polarity does not have a significant effect on the emission color. To further study the different colors in different polymer matrices, the emission of BP-2NT in THF/H₂O mixtures with different water fractions was analyzed to study the effect of aggregation process on luminescence (Figure C6). When a poor solvent of water was added in the THF solution, the emission intensity gradually increased and showed a regular red shift due to the formation of aggregates. So, it hypothesized that the BP-2NT in different polymer matrices could affect the degree of aggregation, and therefore showed different luminescence emissions. The sentence "Due to the photoproduct BP-2NT in different matrices could affect the degree of aggregation, pure BP-2NT film, doped BP-2NT /PMMA film, and doped BP-2NT /PEG film showed different emission colors after light irradiation (Figure S89-91). It means

that the different luminous colors of pattern would also be obtained after the light irradiation.” was added in the revised manuscript and figures C4-C6 were added in the revised supporting information.

Figure C4. Normalized PL spectra of the BP-2NT, BP-2NT/PMMA, and BP-2NT/PEG film. $\lambda_{\text{ex}} = 300$ nm.

Figure C5. Normalized PL spectra of BP-2NT in different polar solutions.

Figure C6. (a) PL spectra of BP-2NT in THF/H₂O solution-aggregates mixtures with different water fractions. (b) Plots of wavelength versus water fractions. Molecular concentration: 10⁻⁵ M; $\lambda_{\text{ex}} = 340$ nm; $I_0 =$ emission intensity in THF ($f_w = 0\%$).

7. The fabrication of 4D information codes should be described in detail, like the concentration of solution, the speed of spin coating, and the size of the glass sheet.

Response: Thanks for the reviewer's careful review. "The fabrication procedure of time-gated 4D codes First, TPE-2MO2NT and polymer stock solutions were prepared by dissolving 1 mg of TPE-2MO2NT in 1 mL CHCl₃ and dissolving 50 mg of the polymer sample in 1 mL of CHCl₃, respectively. Afterward, 0.1 mL of TPE-2MO2NT solution was mixed with 0.2 mL of polymer solution under sonication for about 0.5 h to produce a homogeneous solution doped with the 1.0 wt% TPE-2MO2NT. The films of TPE-2MO2NT doped polymers were fabricated by spin coating the mixed solutions of TPE-2MO2NT and polymer blend onto glass sheets (1.5 cm × 1.5 cm) at 500 rounds/min for 5 s and then 1500 rounds/min for 30 s. The TPE-2MO2NT films were fabricated by spin coating the TPE-2MO2NT stock solutions onto glass sheets (1.5 cm × 1.5 cm) at 500 rounds/min for 5 s and then 1500 rounds/min for 30 s. The obtained films were made in a specific combination to obtain the 4D codes and then irradiated to get different information." was added in the revised supporting information.

8. As TPE-2MO2NT is a new class of cleavage skeleton, the authors can compare with the previous cleavage groups to highlight the advantages of the TPE-2MO2NT compound.

Response: We thank the reviewer for the insightful comment. The sentence "Compared to the common cleavage groups of *o*-nitrobenzene and coumarin^{18, 53}, TPE-2MO2NT/BP-2NT as AIEgens have strong luminescence intensity and high contrast, which could allow for clear tracking and indication of the process change." was added in the revised manuscript.

Response to the Comments of Reviewer 2:

Comments: The only aspect I am able to assess is the synthetic reactivity reported in terms of olefin cleavage since I am not a material scientist.

From that perspective the work in sound and the results obtained are supportive of the proposed mechanism based on singlet oxygen generation and following olefin cleavage. However, this is nothing noteworthy from a synthetic point of view since olefin oxidative cleavage using singlet oxygen can only be achieved on very activated systems like tetra-aryl substituted olefins.

Overall, there is not much of relevance to the synthetic community from this work, which however seems more directed to material science. The impact on this field is something that goes beyond my expertise.

Overall Reply: We thank the reviewer for the comment. Alkenes oxidative cleavage is a common yet important type of chemical reaction in organic synthesis, which usually requires additional catalysts or additives to achieve the break of double bonds (*J. Am. Chem. Soc.* 2021, 143, 10005-10013; *Nature*, 2022, 610, 81-86). While non-catalytic reactions have the advantage of being green, simplified, and low-cost, catalyst-free oxidative cleavage of alkenes has rarely been reported. Fu's group recently reported the catalyst-free alkene photooxidation system for the oxidative cleavage of (Z)-triaryl-substituted alkenes containing a pyridyl motif with configurational dependence (*Green Chem.* 2021, 23, 3649-3655). However, the relationship and extension between the use of catalyst-free alkenes photocleavage reaction and photoresponse processes has not been studied. To further explore the possibilities of catalyst-free alkenes photocleavage reaction for achieving dynamic discoloration in intelligent artificial systems, we constructed the photoresponsive tetraphenylethylene (TPE) derivative TPE-2MO2NT which could undergo efficient catalyst-free photocleavage reaction and accompanied by an obvious blue-shifted fluorescent signal. For one thing, the typical donor-acceptor structure in TPE-2MO2NT favors the self-production of reactive oxygen species (*Adv. Mater.* 2020, 32, 1903530) and promotes the catalyst-free oxidative cleavage of the molecular backbone. For another thing, oxidative cleavage of TPE-2MO2NT compounds breaks the π -conjugation of the molecular backbone, which would significantly influence the electron delocalization of the π -system and thus change the luminescent properties. So, the tetra-aryl substituted olefins containing luminescent units (TPE-2MO2NT) were chosen in our work to realize large wavelength shifts in the photoresponse process, displaying tunable multicolor behaviors from orange to white and finally to blue emission. Combined with the good controllability of photoreaction in different matrixes, dynamic fluorescence images and 4D information codes were further fabricated for advanced information encryption.

The tetra-aryl substituted olefins also could defined as tetraphenylethylenes (TPEs). TPEs are typical aggregation-induced emission luminogens (AIEgens) and have extensive research in the luminous materials field (*Chem. Rev.* 2015, 115, 11718-11940; *Angew.Chem.Int. Ed.* 2020, 59, 9888-9899). It could be utilized as a framework for further elaboration, enabling structure-property-function relationship studies and

multi-functional applications. However, to the best of our knowledge, TPE derivatives with controllable bond cleavage and tunable fluorescence performance has not been reported. Herein, TPE-2MO2NT skeleton with controllable bond breaking with tunable fluorescence performance has been demonstrated for the first time, which is of great potential for applications like photoinduced micropatterns, controlled drug delivery, and photodegradable materials. The new photoresponsive mechanism and molecular skeleton based on the oxidative cleavage of TPE derivatives could open a new avenue for the photoresponsive applications of AIE luminescent materials.

Overall, we construct a new photoresponsive mechanism and molecular scaffold based on the catalyst-free photocleavage reaction of tetraphenylethylene derivatives for advanced information encryption. This work is a good example of the combination of synthetic chemistry and luminescent materials, which provides a new strategy for the design of photofluorochromic luminescent materials and opens a new avenue for the photoresponsive applications of AIE luminescent materials. *Nature Communications* is a multidisciplinary journal that publishes high-quality research from all areas of the natural sciences and aims to represent important advances of significance to specialists within each field. Therefore, we believe this work could be of interest to the readers of *Nature Communications*.

Response to the Comments of Reviewer 3:

Comments: In their manuscript, Lu et al. describe a novel photofluorochromic system based on a highly emissive AIEgen which can be photooxidized selectively under UV radiation to form a different fluorescent species. The authors dutifully characterize the chemical and optical properties of the system and employ DFT methods not only to rationalize the mechanisms but also to explore the properties of a series of similar compounds, identifying the unique properties of TPE-2MO2NT. Finally, they show how it is possible to use the unique features of this system and the color versatility offered by AIE to prepare time-gated photofluorochromic substrates that could find application in anticounterfeiting and cryptography.

The system presented is in itself very interesting and well characterized, as such it can be appealing to broad audiences, spanning both basic and applied fields. However, I find that, in many instances, the study is 'oversold' and the authors claim more novelty than what is actually there. As such I do not believe that the study is suitable for the standard of Nature Communications.

Overall Reply: We greatly appreciate the comment of the reviewer. We have carefully read each of the comments and made extensive revisions to the previous manuscript. We really hope that the response will meet your approval.

Comments: In particular, I believe that is misleading to say that the compound TPE-2MO2NT is by itself responsible for the observed photofluorochromic phenomenon as suggested in the introduction (e.g. lines 103-106, 112-114) and repeated throughout the text (e.g. L161-163). The 'continuous fluorescence change' originates from the disappearing of the AIEgen fluorescence at 580 nm and the appearance of a blue emission at 460 nm ascribable to one of the products of its oxidation. In other words, the continuous fluorescence is due to the synergic effect of two different chemical species, one slowly disappearing and one slowly forming (due to the oxidation of a compound to yield another). This is effectively an irreversible photochromic effect, and, as such, these two contributions should not be regarded only from the perspective of the starting compound (i.e. the photofluorochromisms of TPE-2MO2NT) but in light of the (irreversible) chemical change (i.e. the photofluorochromism of the TPE-2MO2NT/BP-2NT system) which yields a blue fluorophore (as they acknowledge in L196).

Response: We thank reviewer for the comment. We are sorry for the unintentional misleading and we agree with the reviewer that the continuous fluorescence is due to the synergic effect of two different chemical species. We have carefully checked and revised the description of the photofluorochromic phenomenon throughout the manuscript according to the reviewer's suggestions. For details, please refer to the revised materials. Some are listed below:

L32-33, "TPE-2MO2NT displayed tunable multicolor behaviors from orange to white and finally to blue emission with time-dependent" has been revised as "As the original TPE-2MO2NT compound was photocleavage as two new compounds, the system

displayed tunable multicolor behaviors from orange to white and finally to blue emission with time-dependence”.

L103-106, “we designed the first multiple photofluorochromic AIEgen TPE-2MO2NT based on photoinduced oxidative cleavage of alkene” has been revised as “we designed the first multiple photofluorochromic AIE system based on alkene oxidative cleavage photoreaction”.

L112-114, “Furthermore, the C=C bond oxidation will break the original conjugation and thus influence the TICT effect, achieving the continuous fluorescence change with photoirradiation time” has been revised as “The broken of C=C bond will destroy the original conjugation of TPE-2MO2NT to generate the photoconversion system based on BP-2NT and TPE-2MO2NT with tunable multiple fluorescence changes upon light irradiation”.

L161-163, “dynamic photofluorochromic behavior with high-contrast and multiple emission colors within one single molecule was realized by TPE-2MO2NT” has been revised as “dynamic photofluorochromic behavior with high-contrast and multiple emission colors was realized in the TPE-2MO2NT photoconversion process”.

L198, “With the gradual conversion of the orange-emitting TPE-2MO2NT compound to the blue fluorophore BP-2NT, it was concluded that the multicolor photofluorochromic phenomenon is caused by the synergistic effect of TPE-2MO2NT/BP-2NT system.” were added in our revised manuscript.

L451-452, “The photoinduced oxidative cleavage reaction of TPE-2MO2NT was highly efficient and specific, endowing high contrast and tunable multicolor emission” has been revised as “The photoinduced oxidative cleavage reaction of TPE-2MO2NT was highly efficient and specific. As the orange emitting TPE-2MO2NT gradually cleavage to be the blue fluorophore BP-2NT, the whole system was endowed with tunable multicolor emission”.

Comments: In this sense, the system presented here is somewhat different and should not be compared directly to photochromisms phenomena in which the unimolecular nature is obvious (which the author references in ref 10-17 to discuss the state of the art of photofluorochromism) and which are, I believe, more interesting as a result of the fact that more complex colorations can be obtained by rational combination of more entities (see ref 55). The approach the authors decided to follow to describe their system leads to several inconsistencies, such as in L169 and Figure 1e in which the authors argue that the system presents 6 colour changes while having argued before that they obtain a gradual colour shift.

Response: We thank the reviewer for this comment and kind suggestion. As suggested, we have revised the cited references to include some of the photocleavage groups that conversion from the original substance to another compound (*Nat. Commun.* 2021, 12, 2364; *Angew. Chem. Int. Ed.* 2022, 61, e202113163) and also added some reported literatures that showed the synergistic effect of the two photosensitive derivatives (*Adv. Funct. Mater.* 2022, 32, 2107145; *Angew. Chem. Int. Ed.* 2021, 60, 11247-11251).

We fully agree with the reviewer's comment that the elaborate combinations of TPE-2MO₂NT with other compounds enable to obtain more advanced modulated fluorescence systems. Combining multiple molecules to achieve complex applications is a reliable approach in materials science, such as the use of four compounds to achieve multiple color adjustments and applications in ref 55. The development of new reactions and functional groups is the basis and prerequisite for interesting materials and applications, which can bring great application value and space. It should be pointed out that in this work, we mainly focused on developing a new class of photofluorochromic mechanism and molecular scaffold based on the catalyst-free photocleavage reaction. Based on the microenvironmental modulation of the photoreaction process, the preliminary application was studied in dynamic patterning and 4D codes filed. On the basis of this work, the rational combining multiple photofluorochromic TPE-2MO₂NT conversion system with other functional groups to achieve more advanced multicolor fluorescence applications will be an interesting research topic, and we will also carry out related expansion in the future.

In addition, we have unified the description of the photofluorochromic system in the revised manuscript. In view of the fact that the photofluorochromic system exhibits tunable multicolor fluorescence with irradiation time-dependence, we have deleted the expressions like "gradual" and "continuous". The changes we made in the revised manuscript and supporting information are highlighted in red for easy identification.

Comments: The mechanism of the photooxidation is very similar to what was reported in the case of ref 33. (Such mechanism should be added in Scheme 1 for a fair comparison with the state-of-the-art.) While the authors have done an impressive work to investigate the mechanism – combining DFT results, control experiments, and expanding the rationale to structurally related compounds – the latter is not new per se and only applies to a small set of compounds, preventing the generalization to design rules that might guide the synthesis of similar systems.

Response: We thank the reviewer for the comments. According to the reviewer's suggestion, we compared the reaction mechanism and research focus of the ref 33 with ours (Figure C7), and gave relevant explanations in the revised manuscript. Given the intrinsic characteristics of the alkene oxidative cleavage reaction process, achieving catalytic-free cleavage requires the generation of its own reactive oxygen species to break the double bond. Although our work and the ref 33 are based on catalyst-free cleavage reactions of different compound structures, the reaction mechanisms are somewhat similar. In ref 33, catalyst-free oxidative cleavage of (*Z*)-triaryl-substituted alkenes containing pyridyl motif was reported and the mechanism of the reaction is studied systematically. Our work aims to develop new photoresponsive luminogens, elucidate the mechanism of oxidative cleavage photoreaction, and explore the application in advanced information encryption. These are also the major contributions of our work and very different from the ref 33 work, which focus on the organic chemistry field. Our work is a good example of the combination of synthetic chemistry and luminescent materials, which provides a new strategy for the design of photofluorochromic luminescent materials and opens a new avenue for the

photoresponsive applications of AIE luminescent materials. The sentences “In contrast to other catalytic-free oxidative cleavage system focusing on the organic synthesis aspect³⁴, this study systematically investigated the photoreaction mechanism and for the first time applied the photoreaction to construct the photoresponsive mechanism and obtain a new photocleavage skeleton (Figure S28).” were added in our revised manuscript.

Figure C7. Comparison of reaction mechanism and research focus of (a) our work and (b) previously reported work.

To better understand the photochemical processes of TPE derivatives, different electron-donating and electron-withdrawing groups were decorated on the TPE skeleton to get seven compared compounds for study of their photoresponsive behavior in our work. Compared with the efficient photocleavage of TPE-2MO2NT, TPE-2M2NT and TPE-2MO2CN could slowly undergo the oxidative photocleavage reaction. TPE-4MO rapidly underwent photocyclization reaction under light, while TPE-2M2Ph and TPE-2MO2AB mainly underwent the photocyclization reaction at a slow rate. TPE-2M2NB and TPE-4NT had no obvious photoreaction. The results show that rapid photocleavage, slow photocleavage, rapid photocyclization, slow photocyclization, and photostabilization of TPE derivatives can be regulated. Since TPE derivatives are extensively used in the luminescence field and their photoresponse properties have also been developed, our results could provide useful information for the development and design of other TPE photoresponsive compounds. The sentences “The results showed that the photocleavage, photocyclization, and photostabilization of TPE derivatives can be regulated by different substituents, which also could provide useful information for the photoresponsive design of other TPE compounds.” were added in our revised manuscript.

Through the systematic study of the reaction mechanism, it is proved that the cleavage photoreaction requires a specific chemical structure, and not all

tetraphenylethylene compounds can efficiently undergo this reaction. It indicates the design of TPE-2MO₂NT skeleton is unique and impactful. To illustrate the universality of TPE-2MO₂NT skeleton, three new luminogens were also synthesized in our work. It can be inferred that the TPE-2MO₂NT skeleton can be introduced into small molecules or polymer systems to realize the photocleavage change. For example, we are recently investigating the introduction of TPE-2MO₂NT skeleton into the polymer backbone to realize the degradation of polymers and monomer recycling under light irradiation. TPE-2MO₂NT skeleton has the advantage of controllable bond broken with tunable fluorescence performance, which is of great potential for applications like photoinduced micropatterns, controlled drug delivery, and photodegradable materials. We believe that this work could serve as an inspiring starting point for bringing up some new discoveries and applications in the future. The sentences “The new photocleavable luminescent skeleton and its cleavage photoreaction could provide a new starting point for future photoresponsive application possibilities, like photoinduced micropatterns, controlled drug delivery, and photodegradable materials.” were added in our revised manuscript.

Comments: Finally, compared to other previous works of the authors (e.g., 10.1002/anie.202208460, 10.1038/s41467-023-38801-1, 10.1002/adfm.202213927) and those referenced in the study, I found the proposed anti-counterfeiting application rather underwhelming. While I agree with the rationale behind the time-evolving codes, the one proposed by the authors and described in Figure 5 suffers some drawbacks. For example, the readout input is the same as the stimulus that triggers the change and as such, unless the only true information will be the last code generated, some information (in Fig 5 the authors identify code 2 also as a true info) can only be read for a short period of time, after which is irreversibly lost. In addition, it takes several minutes to get to the final (true) configuration so information retrieval will be quite slow and not functional. On top of that, once such configuration is reached, as the fluorescence won't change anymore, looking through UV light will reveal the right info and in this sense the cryptographic key is only one-use.

Response: We thank the reviewer for comment. The anti-counterfeiting applications in the three papers are excellent, and we added relevant references in the applications section. The applications in the three papers are based on the photoresponsive groups and photoreactions that have been reported, and have different selling points from our work. We mainly focus on the development of new photoresponsive skeleton and exploration of the mechanism based on oxidative cleavage photoreaction. It provides a design strategy to explore catalyst-free photoreaction for designing the new photoresponsive skeleton. By investigating photoreactions in matrices with multicolor microenvironmental controllability, dynamic images and 4D information codes were constructed. Different information codes can be obtained under light exposure, and true information and wrong information can be defined according to the application requirements. The development of new reactions and functional groups is the basis and prerequisite for interesting materials and applications, which can bring great application value and space. Since the TPE-2MO₂NT skeleton and its photoreaction have the

advantage of controllable bond breaking with tunable fluorescence performance, it is of great potential for applications like photoinduced micropatterns, controlled drug delivery, and photodegradable materials. As a proof of concept, the application in our work is a prototype demonstration, which can be further improved and extended in future studies.

Furthermore, a range of parameters such as light intensity, emission wavelength, and exposure distance can be adjusted precisely to control photoreaction processes. Therefore, the time to obtain information can be controlled by controlling the intensity of the light. For example, the photoreaction process can be accelerated by increasing the power of the light source to shorten the time to acquire information. In addition, the information in the middle state could disappear after reading, which has the potential for fabricating transient information storage materials for Snapchat (disappear after reading) against rising problems in counterfeiting (*Angew. Chem. Int. Ed.*, 2021, 60, 3640-3646; *Angew. Chem. Int. Ed.*, 2023, 135, e202313728). Based on our application, it can be judged that more complex information encryption systems could be obtained by regulating a variety of external parameters to fulfill the different application requirements. The sentence “Since the photoreaction process can be controlled by parameters, such as light intensity, emission wavelength, and exposure distance. Therefore, the time for acquiring information by irradiation of the 4D code can be changed. For example, by increasing the light intensity to shorten the time for obtaining information. It can be judged that with the adjustment of the parameters, the 4D information anti-counterfeiting system can be improved.” were added in our revised manuscript.

We again thank the reviewers very much for the constructive suggestions and comments, which have been fully addressed in the revised manuscript. We hope the manuscript has been improved a lot and the revised manuscript does meet the criteria for publication in *Nature Communication*.

REVIEWERS' COMMENTS

Reviewer #1 (Remarks to the Author):

The revised manuscript has comprehensively addressed my comments. I recommended the acceptance in current copy.

Reviewer #3 (Remarks to the Author):

The revisions improved the content and the clarity of the manuscript. The authors addressed most of the concerns of the Reviewers but I believe some issues still remains concerning the claimed novelty of the synthetic effort and its scope, as also brought up by the other Reviewers. This is quite relevant, in my opinion, considering the importance that is given to this aspect in the manuscript, e.g. the first image is about the synthetic novelty.

In light of what the authors discussed about ref 34 (previously ref 33) and the image they added as S28, the novelty that the authors claim in Scheme 1 is not consistent. The latter should be updated to include the reaction scheme of S28 as 'Previous works' and the 'novelty ticks' changed accordingly (i.e., 'ROS generation Reactive oxidant' and 'Activated alkene Easily be oxidized' should be removed).

In the rebuttal and in the added part of the text the authors mention how TPE-2MO2NT can be regarded as a new photocleavable skeleton yet they also mention its unique properties compared to other push-pull TPE systems. How can the authors be sure that the chemical modifications necessary to make the use of such compound in a more general way (i.e. introducing solubilizing side groups or covalently attach TPE-2MO2NT to a material) won't change its behaviour?

Responses to the Comments and Suggestions of Reviewers

Response to the Comments of Reviewer 1:

Comments: The revised manuscript has comprehensively addressed my comments. I recommended the acceptance in current copy.

Reply: We appreciate the reviewer affirmation of our research and constructive suggestions for improving our manuscript.

Response to the Comments of Reviewer 3:

Comments: The revisions improved the content and the clarity of the manuscript. The authors addressed most of the concerns of the Reviewers but I believe some issues still remains concerning the claimed novelty of the synthetic effort and its scope, as also brought up by the other Reviewers. This is quite relevant, in my opinion, considering the importance that is given to this aspect in the manuscript, e.g. the first image is about the synthetic novelty. In light of what the authors discussed about ref 34 (previously ref 33) and the image they added as S28, the novelty that the authors claim in Scheme 1 is not consistent. The latter should be updated to include the reaction scheme of S28 as 'Previous works' and the 'novelty ticks' changed accordingly (i.e., 'ROS generation Reactive oxidant' and 'Activated alkene Easily be oxidized' should be removed).

Reply: We thank the reviewer for the comment. According to the reviewer's suggestion, we revised the Figure 1 (previously Scheme 1) and its caption in the revised manuscript. Alkenes oxidative cleavage is an important type of chemical reaction in organic synthesis and basically requires additional catalysts or additives to achieve the break of double bonds. So far, only one work (ref 34) reported the catalyst-free alkene photooxidation system (Green Chem. 2021, 23, 3649-3655). It is essential to develop more catalyst-free system and study the photophysical properties of alkenes oxidative cleavage reaction. For the accuracy of the description, the drawback of previous works and the novelty ticks of this work were modified in Figure 1.

Figure 1. Examples of oxidative cleavage of alkenes in reported works and the catalyst-free oxidation of TPE-2MO2NT in this work. TPE-2MO2NT is modified by two electron-donating methoxy groups and two electron-withdrawing naphthalimide units, producing two photoproducts BP-2NT and BP-2MO. AIE: aggregation-induced emission; TICT: twisted intramolecular charge transfer; ROS: reactive oxygen species.

Comments: In the rebuttal and in the added part of the text the authors mention how TPE-2MO2NT can be regarded as a new photocleavable skeleton yet they also mention its unique properties compared to other push-pull TPE systems. How can the authors be sure that the chemical modifications necessary to make the use of such compound in a more general way (i.e. introducing solubilizing side groups or covalently attach TPE-2MO2NT to a material) won't change its behaviour?

Reply: We thank the reviewer for the comment. Three derivatives of TPE-2MO2NT, TPE-MN-C5, TPE-MN-Br, and TPE-MN-TA modified with alkyl chains and ionic groups, have demonstrated photocleavage properties under light irradiation. Given the intrinsic characteristics of the alkene oxidative cleavage reaction process, achieving catalytic-free cleavage requires the generation of its own reactive oxygen species (ROS) to break the double bond. We speculate that for further substituent modifications of TPE-2MO2NT, it is necessary to ensure adequate generation of ROS and enable the double bond moiety as the most susceptible site to oxidation within the compound structure. We will further study the correlation between TPE-2MO2NT photocleavable structure and performance in the future. The sentences “To enrich the scope and applications of TPE-2MO2NT photocleavable skeleton, it is worth studying and continuing to focus on the introduction of appropriate substituents to modify TPE-2MO2NT in the future.” were added in our revised manuscript.